# Efficacy Evaluation of an Intradermally Delivered Enterotoxigenic *Escherichia coli* CF Antigen I Fimbrial Tip Adhesin Vaccine Coadministered with Heat-Labile Enterotoxin with LT(R192G) against Experimental Challenge with Enterotoxigenic *E. coli* H10407 in Healthy Adult Volunteers

**DOI:** 10.3390/microorganisms12020288

**Published:** 2024-01-29

**Authors:** Ramiro L. Gutiérrez, Chad K. Porter, Clayton Harro, Kawsar Talaat, Mark S. Riddle, Barbara DeNearing, Jessica Brubaker, Milton Maciel, Renee M. Laird, Steven Poole, Subra Chakraborty, Nicole Maier, David A. Sack, Stephen J. Savarino

**Affiliations:** 1Naval Medical Research Command, Silver Spring, MD 20910, USA; rlgm72@gmail.com (R.L.G.); markriddlemd@gmail.com (M.S.R.); reneemlaird@gmail.com (R.M.L.); spoole102@gmail.com (S.P.); s.j.savarino@gmail.com (S.J.S.); 2Center for Immunization Research, Johns Hopkins Bloomberg School of Public Health, Baltimore, MD 21224, USAktalaat@jhu.edu (K.T.); bdenear1@jhu.edu (B.D.); dsack1@jhu.edu (D.A.S.); 3Henry M. Jackson Foundation for the Advancement of Military Medicine, Bethesda, MD 20817, USA; 4Department of International Health, Johns Hopkins Bloomberg School of Public Health, Baltimore, MD 21205, USA; schakr1@jhu.edu; 5PATH, Washington, DC 20001, USA; nmaier@path.org

**Keywords:** enterotoxigenic *E. coli*, CfaE, intradermal, vaccine, efficacy, H10407, LTR192G antigen/adjuvant

## Abstract

Background. Enterotoxigenic *E. coli* (ETEC) is a principal cause of diarrhea in travelers, deployed military personnel, and children living in low to middle-income countries. ETEC expresses a variety of virulence factors including colonization factors (CF) that facilitate adherence to the intestinal mucosa. We assessed the protective efficacy of a tip-localized subunit of CF antigen I (CFA/I), CfaE, delivered intradermally with the mutant *E. coli* heat-labile enterotoxin, LTR192G, in a controlled human infection model (CHIM). Methods. Three cohorts of healthy adult subjects were enrolled and given three doses of 25 μg CfaE + 100 ng LTR192G vaccine intradermally at 3-week intervals. Approximately 28 days after the last vaccination, vaccinated and unvaccinated subjects were admitted as inpatients and challenged with approximately 2 × 10^7^ cfu of CFA/I+ ETEC strain H10407 following an overnight fast. Subjects were assessed for moderate-to-severe diarrhea for 5 days post-challenge. Results. A total of 52 volunteers received all three vaccinations; 41 vaccinated and 43 unvaccinated subjects were challenged and assessed for moderate-to-severe diarrhea. Naïve attack rates varied from 45.5% to 64.7% across the cohorts yielding an overall efficacy estimate of 27.8% (95% confidence intervals: −7.5–51.6%). In addition to reducing moderate–severe diarrhea rates, the vaccine significantly reduced loose stool output and overall ETEC disease severity. Conclusions. This is the first study to demonstrate protection against ETEC challenge after intradermal vaccination with an ETEC adhesin. Further examination of the challenge methodology is necessary to address the variability in naïve attack rate observed among the three cohorts in the present study.

## 1. Introduction

Enterotoxigenic *E. coli* (ETEC) is a leading cause of diarrhea in children in low to middle-income countries, travelers, and deployed military in those regions [1]. Such infections are associated with acute impacts, as well as long-term physical and cognitive health deficits in children [2], and long-term chronic functional bowel diseases and other sequelae in adults [3,4], potentially affecting mission readiness in military personnel [5]. The development of improved prevention and control measures for ETEC remains a WHO priority, and international public health stakeholders have recently urged accelerated vaccine development given the increasing prevalence of antimicrobial resistance [6,7,8].

Despite the clear need, there are no licensed vaccines. One of the impediments to vaccine development is that ETEC expresses multiple virulence factors including over 25 different colonization factors (CF), heat-labile toxin (LT), and/or heat-stabile toxin (STp and STh). This phenotypic diversity has impeded the vaccine development process. While many current vaccine approaches target common CFs along with LT, CF diversity and the need to develop a broadly protective vaccine require careful consideration and antigen prioritization [9]. In addition, strategies to overcome suboptimal immune responses in oral vaccinations in certain populations necessitate evaluations of parenteral routes capable of protecting against a mucosal pathogen [8].

One of the most well-characterized antigens is the class 5 CF, CFA/I, a rigid rod-like fimbria with a tip adhesin protein, CfaE, that is thought to facilitate the intestinal adhesion of the bacterium [10,11]. Animal studies indicate that CfaE contains conserved epitopes that might induce broader protection against ETEC strains sharing these epitopes [12]. We have previously developed a stable monomeric form of CfaE (dscCfaE) [10] that is highly immunogenic when co-administered with an LT-based antigen in mice and *Aotus nancymaae* by the intranasal (IN) [13] and intradermal (ID) [14] routes and in mice by the transcutaneous route [15]. This vaccine was protective in both the infant mouse and *A. nancymaae* CFA/I+ H10407 challenge models [16,17], and anti-dscCfaE hyperimmune bovine colostrum passively protects human subjects against H10407 challenge [18]. Subsequently, the safety and immunogenicity of dscCfaE coadministered intradermally with LTR192G was evaluated in an open-label dose-escalating Phase 1 trial [19]. Given the safety profile of the vaccine and the robust anti-CfaE and LTB serum and mucosal immune responses, we conducted a preliminary assessment of the vaccine’s efficacy using a controlled human infection model (CHIM) with ETEC strain H10407.

## 2. Methods

### 2.1. Clinical Trial Design

Healthy US adult volunteers were vaccinated intradermally in three cohorts with 25 μg dscCfaE + 100 ng LTR192G on study days 0, 21, and 42 (Figure 1). No placebo vaccinations were utilized. Vaccinated subjects were eligible for challenge 28 days after receiving their third vaccination if they continued to meet the inclusion and exclusion criteria. Unvaccinated, naïve subjects were recruited for challenge to confirm challenge attack rates and to enable estimates of preliminary efficacy. Though originally intended to be a two-cohort study, discrepancies in attack rates in naïve subjects between the initial two cohorts, as delineated below, led to the addition of a third cohort. This Phase 2b vaccination-challenge study was approved by the Naval Medical Research Command (NMRC.2013.0011) and the Western Institutional Review Boards (WIRB PRO NUM: 20130827) in compliance with all Federal regulations governing the protection of human subjects and was registered in ClinicalTrials.gov (ClinicalTrials.gov Identifier: NCT01922856).

### 2.2. Study Population and Enrollment Criteria

Volunteers were healthy male and non-pregnant female adults between 18 and 50 years of age recruited from throughout the Eastern seaboard. Subjects had to provide written informed consent and pass a comprehension assessment. Subjects were excluded if they had a significant medical or psychiatric condition or had clinical laboratory abnormalities. Subjects with a positive blood test for Hepatitis B surface antigen, Hepatitis C virus, or Human Immunodeficiency Virus-1 were also excluded. Evidence of IgA deficiency was also exclusionary. Due to the inpatient nature of the CHIM, evidence of current alcohol or drug dependence was exclusionary as was recent vaccination or receipt of another investigational product (≤30 days).

To limit enrollment to naïve subjects, volunteers with potential ETEC exposures in the past 3 years based on self-reported medical, travel, and occupational history were excluded. Further, volunteers were excluded if they regularly used anti-diarrheal, anti-constipation, or antacid therapy, had an abnormal stool pattern (<3 stools per week, or >3 stools per day), used a medication known to affect the immune function prior to vaccination (≤30 days), or had a known allergy to 2 of the following antibiotics: ciprofloxacin, trimethoprim-sulfamethoxazole, and penicillin. Lastly, given the intradermal route of vaccination, volunteers were excluded if they had current acute skin infections, severe acne, active contact dermatitis, or a past/current medical history of chronic skin disease or any skin findings that would inhibit adverse event (AE) monitoring or increase AE risk. This study utilized a screening protocol to identify eligible subjects, and only those who were initially eligible underwent per-protocol screening.

### 2.3. Immunization Procedures

After screening and enrollment, eligible subjects were vaccinated ID in alternating upper arms on days 0, 21, and 42 (Figure 1) as described elsewhere [19]. CfaE and LTR192G doses were verified post-vaccination by SDS-PAGE and Western blot analyses. Immediately post-dosing, subjects were assessed for evidence of wheals, blebs, and flashback as well as being asked to rate their vaccination site pain. Subjects were observed for 30 min post-vaccination and vital signs were collected. Memory aid tools were given to record local and systemic reactions. Following the first vaccination, subjects returned one and 7 days after application for evaluation and assessment of adverse events. These clinical evaluations were repeated 7 days after the second and third vaccinations, and a follow-up phone call was made on day 62 (±2 days) to review adverse events.

All vaccine site rash/skin lesions were described using a vaccine site grading scale (VSAGS) designed to standardize the description of the appearance of the rashes [20]. The VSAGS is not a severity grading scale but rather a methodology instituted to standardize the description of rash/skin lesions. Study clinicians evaluated any rash/skin lesion that developed using this scale. This was a separate assessment from the local AE assessment. In accordance with the scale, the site was assessed for erythema, induration, edema, ecchymosis, papules/plaques, vesicle/bullae, hyperpigmentation, and hypopigmentation.

### 2.4. Manufacture and Analysis of cGMP dscCfaE and LTR192G Vaccine Components

The investigational vaccine components, dscCfaE and LTR192G, were manufactured using current Good Manufacturing Practices (cGMP) and are described elsewhere [19].

### 2.5. Bacterial Challenge Strain Preparation

The ETEC strain was H10407 (LT^+^ ST^+^ CFA/I^+^) [21,22,23]. The inoculum was freshly prepared from vials (Lot #0519) produced under cGMP conditions at the Walter Reed Army Institute of Research Pilot Bioproduction Facility. H10407 was grown on CFA agar with bile salts overnight at 37 °C and harvested in sterile phosphate-buffered saline (PBS). The final concentration of colony-forming units (cfu) was determined by optical density and confirmed by plate count.

### 2.6. Challenge Procedures

Subjects were admitted to the inpatient facility at the Center for Immunization Research at the JHU Bayview Campus the day prior to challenge and evaluated for continued eligibility and had baseline samples collected. All subjects initiated an overnight fast and were challenged the next day with a target inoculum of 2 × 10^7^ CFU of ETEC strain H10407 [21,22,23,24]. Subjects were monitored for diarrhea and other signs of enteric illness. Subjects were treated with ciprofloxacin (500 mg by mouth twice daily for three days) five days after challenge, or sooner if they met early treatment criteria. This study was designed to be conducted over a series of sequential cohorts due to the capacity of the inpatient facility.

### 2.7. Antigen-Specific ELISA

Anti-dscCfaE and anti–LTB IgG and IgA antibody (Ab) titers were quantified by ELISA in serum samples obtained on days 0, 21, 42, 69, and 98. IgG assays: 96-well Nunc™ MicroWell™ microplates (Thermo Scientific, Rochester, NY, USA) were coated with 100 µL/well of dscCfaE at 1 µg/mL or GM1 (Sigma-Aldrich, Saint Louis, MO, USA) at 0.5 µg/mL (for LTB-specific assays), for 1 h at 37 °C, followed by overnight (ON) at 4 °C. Blocking was performed with 200 μl/well of 5% non-fat milk (Sigma-Aldrich) in 0.05% Tween-20 (Sigma-Aldrich)-PBS (PBS-T) for 60 min/37 °C in a humidified chamber. LTB-specific plates were added of LTB at 0.5 µg/mL and incubated at 37 °C for 1 h. After three washes with PBS-T, serum samples were added at a starting dilution of 1:50 in 1% non-fat milk-PBS-T followed by a 3-fold serial dilution and incubated for 1.5 h at 37 °C in a humidified chamber. After five washes with PBS-T, peroxidase-conjugated goat anti-human IgG (KPL, Gaithersburg, MD, USA) was added at 0.5 µg/mL in 1% non-fat milk-PBS-T for 1.5 h at room temperature (RT). After final washes, 2,2′-azino-bis(3-ethylbenzothiazoline-6-sulphonic acid (ABTS) (KPL) was added according to the manufacturer’s recommendations. After 30 min incubation, optical density (OD) was measured at 450 using a Multiskan EX® ELISA reader with Ascent® software (Thermo Scientific). IgA assays: 96-well Nunc™ Maxisorb™ microplates (Thermo Scientific) were coated with 100 µL/well of dscCfaE at 1 µg/mL or GM1 (Sigma-Aldrich) at 0.5 µg/mL (for LTB-specific assays), for 1 h at 37 °C, followed by ON at 4 °C. Plates were blocked with 200 μL/well of 5% non-fat milk (Sigma-Aldrich) in 0.05% PBS-T for 60 min at 37 °C in a humidified chamber. LTB was added to LTB-specific plates at 0.5 µg/mL and incubated at 37 °C for 1 h. After three washes with PBS-T, serum samples were added at 1:50 in 2% non-fat milk-PBS-T followed by a 3-fold serial dilution and incubated for 1.5 h at 37 °C in a humidified chamber. Plates were washed five times with PBS-T followed by addition of 0.25 µg/mL of biotin-conjugated anti-human IgA (KPL) in 2% non-fat milk-PBS-T for 1.5 h at RT. After further washes, ExtrAvidin®-Peroxidase (Sigma-Aldrich) was added at 1:2000 for 30 min at RT. After final washes, 3,3′,5,5′-tetramethylbenzidine (Ultra-TMB) (Thermo Scientific) was added according to the manufacturer’s recommendations. After 30 min incubation, OD was measured at 405 nm. The cutoff for each plate was calculated by the average of the background wells’ OD plus a fixed value of 0.4. A linear regression was fitted to the experimental data, and the endpoint titer was determined as the reciprocal of the interpolated sample dilution that intersected with the cutoff. The final results were calculated as the average of the duplicate assays. Samples with OD below the cutoff were assigned a value of one-half the lower dilution tested (i.e., 1:25). Titers were log10-transformed for statistical analyses and seroconversion was defined as a 4-fold increase over baseline titer.

### 2.8. Antibody in Lymphocyte Supernatant (ALS)

Peripheral Blood Mononuclear Cells (PBMCs) isolated by Ficoll–Hypaque gradient on days 0, 7, 28, 49, 69, and 77 were used to quantify the levels of anti-CfaE and -LTB IgG and IgA antibodies (antibody in supernatant, ALS), as an indirect measurement of antigen-specific antibody-secreting cells. Fresh isolated PBMC were resuspended in complete RPMI (10% heat-inactivated fetal calf serum, 1% Penicillin-Streptomycin (Life Technologies, Carlsbad, CA, USA) and 1% GlutaMAX™ (Life Technologies), and incubated in duplicate in 24-well plates (Corning Inc., Corning, NY, USA) at 5 × 10^6^ cells/mL, 1 mL/well, at 37 °C and 5% CO_2_ for 72 h. Following incubation, culture supernatants were collected and stored at −80 °C until tested by ELISA as described above. Responders were defined as individuals with a 4-fold increase in baseline titer.

### 2.9. Primary (and Secondary) Study Endpoints and Definitions

The primary endpoint was moderate-to-severe diarrhea according to the following definitions: (1) severe diarrhea: ≥6 grade 3–5 stools in 24 h, or >800 g of grade 3–5 stools in 24 h; (2) moderate diarrhea: 4–5 grade 3–5 stools in 24 h or 401–800 g of grade 3–5 stools in 24 h. Secondary endpoints included time to onset and duration of diarrhea, total number and volume of loose stools, maximum 24 h stool output, percent of subjects with severe diarrhea or diarrhea of any kind, percent of subjects with nausea, vomiting, anorexia, or abdominal pain/cramps rated as moderate to severe, number of cfu of H10407 per gram of stool 2 and 4 days post-challenge and ETEC systemic and diarrhea severity score post-challenge with H10407 [25].

### 2.10. Outcome Adjudication

This study was designed as an open-label trial due to the 100% local reactogenicity rates in vaccine recipients from the Phase 1 trial [19]. In an effort to obtain an unbiased determination of the efficacy outcomes, an independent outcome adjudication committee was utilized as has been previously described for other challenge trials [26]. Briefly, independent researchers blinded to vaccination status evaluated challenge outcome data after study to make an independent assessment of the primary endpoint.

## 3. Data Analysis and Statistical Considerations

Approximately 40 subjects were originally targeted for vaccination of whom 28 were planned for challenge concurrently with 28 unvaccinated subjects. The null hypothesis was that the percentage of subjects with moderate-to-severe diarrhea would be the same in vaccinated and unvaccinated subjects. Assuming a >70% moderate-to-severe diarrhea rate in unvaccinated subjects and an attack rate of <30% in vaccinated subjects (equivalent to >57% protective efficacy), a total of 28 subjects per group yielded an 80% power to detect a significant difference in attack rates (2-sided alpha = 0.05). After challenging 56 subjects, an additional challenge cohort was enrolled due to aberrantly low attack rates in cohort 1. This interim analysis required an alpha adjustment using the Pocock method [27] and indicated that a total of 54 subjects (23 vaccine and 31 placebo recipients) yielded an approximate 80% power with an alpha = 0.0294. This updated analysis was anticipated to focus on only cohorts in which the naïve subjects had an attack rate of 58–82%, as that was more reflective of the anticipated attack rate with a 2 × 10^7^ cfu dose of H10407 following an overnight fast [24].

Vaccine efficacy was calculated as 1 minus the relative risk of moderate-to-severe diarrhea rates among controls and vaccine recipients and reported with 95% confidence intervals. Estimates of the common relative risk by study group across cohorts were calculated with the Cochran–Mantel–Haenszel method [28,29]. The Breslow–Day test was used to assess whether the vaccine’s effect (expressed as the odds ratio) was consistent across strata [30]. Other statistical comparisons for nominal outcomes were made using Pearson’s chi-square test or Fisher’s exact test. An ETEC Disease Severity Score as described by Porter et al. was utilized to evaluate the combined effect of vaccination on all ETEC-attributable symptoms [25]. Continuous endpoints (e.g., log_10_ endpoint titers, stool output, etc.) were compared by *t*-tests or Wilcoxon tests, depending on whether assumptions were fulfilled. All statistical tests were interpreted in a two-tailed fashion using alpha = 0.05.

## 4. Results

### 4.1. Subjects

Among the 56 subjects who signed a written informed consent and underwent study-specific screening for the vaccination phase of the study, 52 received all three vaccine doses (Figure 1). The initial cohort of subjects consisted of 23 vaccinated subjects originally enrolled and receiving an initial vaccination. One received only two vaccinations due to medical concerns unrelated to study participation. Of the 23 vaccinated subjects, 19 proceeded to the challenge phase and were challenged with H10407 (1.0 × 10^7^ cfu) along with 11 naïve subjects. The next cohort consisted of 14 vaccinated subjects, one of whom only received a single vaccination due to a family emergency. A total of 9 vaccinees advanced to challenge concurrently with 17 naïve subjects (1.2 × 10^7^ cfu). The last cohort included 19 vaccinated subjects, two of whom only received two doses due to new jobs precluding their continued participation in the final vaccination or challenge. Of the vaccinated subjects, 13 joined 15 naïve subjects and were challenged concurrently (1.9 × 10^7^ cfu). The mean age of participants was 34.4 years (SD 8.3) and the majority were male (64.3%) and black (86.9%). Demographic characteristics were comparable across the three cohorts and study groups (Table 1).

### 4.2. Vaccine Safety

There were no serious adverse events associated with the vaccine or any adverse events that led to subject withdrawal, and all AEs observed were consistent with those anticipated with the administered vaccine and indicated no safety concerns. Table 2 details the surveyed signs and symptoms of subjects receiving at least one vaccine dose. All subjects experienced a vaccine site reaction (*n* = 56, 100%), and most also experienced vaccine site pruritis (*n* = 49; 87.5%) and/or tenderness (*n* = 32; 57.1%). Vaccine site pain (*n* = 11; 19.6%) and swelling (*n* = 8; 12.5%) were also observed in a small proportion of vaccinees. In terms of vaccine site appearance, all reactions were at some point during their course characterized as erythematous (Table 3). Hyperpigmentation and induration were also present in 100% of vaccinated subjects and were often noted upon resolution of erythema. Six months after initial vaccination, 16 subjects (28.6%) reported persistent hyperpigmentation and 1 (1.8%) reported hypopigmentation. At the twelve-month follow-up, this had decreased to 5.4% and 0.0%, respectively.

Systemic AEs following vaccination were uncommon and included headache (*n* = 8; 14.3%), malaise (*n* = 4; 7.1%), arthralgia (*n* = 4; 7.1%), and chills (*n* = 3; 5.4%). There were no chronic illnesses or serious health conditions reported by any individuals contacted at the 6- or 12-month follow-up.

### 4.3. Vaccine Efficacy

A total of 41 vaccine recipients and 43 naïve subjects advanced to the challenge phase of the study. Table 4 shows the challenge doses and rates of moderate-to-severe diarrhea (MSD) administered over the three separate study cohorts. The overall MSD rate among naïve subjects was 55.8% (24/43) compared to 39.0% (16/41) among vaccine recipients, yielding a combined efficacy estimate of 27.8% (95% confidence intervals: −7.5–51.6%).

Across all cohorts, vaccination was associated with milder disease in terms of non-diarrheal outcomes such as abdominal cramps, pain, and vomiting (Table 5). Additionally, the mean loose stool output among vaccine recipients was lower than in naïve subjects both as a measure of total output (631 mL and 1446 mL, respectively; *p* = 0.03) as well as maximum 24 h output (475 mL and 1037 mL, respectively; *p* = 0.02). This was also a clear reduction in the mean loose stool output in the 120 h post-challenge (Figure 2). Finally, ETEC disease severity was significantly lower (*p* = 0.02) in vaccine recipients (median: 2; interquartile range: 1–5) compared to naïve subjects (median: 3; interquartile range: 1–7) (Figure 3) which corresponded to increasing vaccine efficacy with increasing disease severity (Figure 4).

### 4.4. Immune Responses

Among vaccinees, all subjects (100%) had at least a four-fold increase in serum IgG Ab titers to CfaE (Table 6 and Figure 5a), and the magnitude of response appeared to increase throughout the vaccination series. Serum IgA responses to CfaE were lower (61.0%; *n* = 25) with a lower magnitude of response and appeared to plateau after the second vaccination (Figure 5b). Anti-LTB responses also appeared to peak prior to receipt of the third vaccination with observed IgG response rates of 87.8% (*n* = 26) and IgA response rates of 63.4% (*n* = 26). During the challenge period, antibody (IgG and IgA) titers to both antigens remained approximately the same as pre-challenge titers among vaccine recipients (Figure 5c,d). Among naïve subjects, anti-CfaE titers were also relatively unchanged with only a slight, non-significant increase in IgA titers. Anti-LTB IgG and IgA Ab titers appeared to increase post-challenge in naïve participants; however, these increases were not statistically significant.

ALS responses peaked in vaccinees 7 days after receipt of the second vaccination with response rates of 97.6% (*n* = 40) and 68.3% (*n* = 28) to CfaE and LTB, respectively (Table 6 and Figure 6). Naïve subjects demonstrated significant increases in anti-CfaE and anti-LTB IgA ALS titers (*p* < 0.001) following H10407 challenge.

## 5. Discussion

Here, we confirmed the safety of intradermally administered dscCfaE + LTR192G observed in the prior Phase 1 trial [19]. The vaccine was well-tolerated, and local vaccine site reactions were common but of relatively mild severity. Furthermore, immune responses to both vaccine components were robust and followed similar patterns as the Phase 1 trial [19].

The dscCfaE + LTR192G vaccine did not elicit a statistically significant reduction in moderate-to-severe diarrhea rates in vaccinated participants compared to contemporaneously challenged naïve subjects. As displayed in Table 4, the vaccine efficacy confidence intervals for cohort 2 do not cross 0%; however, across all three cohorts, there was heterogeneity that was borderline statistically significant (*p* = 0.095), indicating that the combined efficacy estimate should be interpreted with caution due to heterogeneity in the efficacy estimates across the three cohorts. Despite not meeting our primary efficacy endpoint, vaccination did significantly reduce the ETEC disease with a reduction in the overall ETEC disease severity score, lower loose stool output, and a lower frequency of ETEC-attributable signs and symptoms. These findings build upon our prior work that demonstrated that passively administered bovine colostrum antibodies against CfaE prevent ETEC-mediated disease following challenge with the CFA/I-expressing ETEC strain H10407 [18]. Additionally, we saw an increased efficacy of the vaccine against more severe disease endpoints, similar to what was reported previously with a candidate *Shigella* bioconjugate vaccine [31]. Such data support the use of clinical outcomes that better capture the full spectrum and severity of disease as opposed to the primary focus on diarrhea volume and frequency. Collectively, these results demonstrate the ability to reduce ETEC illness severity using parenteral immunization with an adhesin-based subunit vaccine and support the further evaluation of other key adhesin antigens to expand vaccine coverage.

We saw significant heterogeneity in moderate-to-severe diarrhea rates across inpatient cohorts among naïve (and vaccinated) participants, with aberrantly low MSD rates seen in the naïve participants in cohort 1 and to a lesser extent in cohort 3 (Table 4). Only cohort 2 yielded a naïve attack rate consistent with prior studies using the H10407 strain at this dose following overnight fasting [24]. This variability may have contributed to an inability to detect a significant difference between vaccinees and naïve subjects. Efforts to identify the potential underlying reason for the heterogeneity, including baseline anti-ETEC antibody titers, were unsuccessful. Though baseline serum anti-LTB IgG and IgA titers were higher than anticipated in challenged naïve subjects, these titers were not significantly different than at baseline in vaccinated volunteers, and there was no correlation between anti-LTB titers and challenge outcome. Additionally, challenged naïve subjects mounted an anti-CfaE and -LTB IgA ALS response indicating mucosal exposure to H10407 induced robust mucosal antibody responses despite heterogeneity in the attack rate. Importantly, the data highlight the challenges associated with enteric CHIMs and point to the need for their continued refinement and standardization. For example, McArthur et al. recently reported on the utilization of a frozen H10407 inoculum as one means to standardize the model [32]. It is unknown if the utilization of such an inoculum for our study would have yielded more consistent disease rates across cohorts. Similar strategies have been utilized for *S. sonnei*; however, more studies are needed to ascertain whether this has yielded consistent disease rates over time and study [33,34].

While there was not a significant reduction in the rates of moderate-to-severe diarrhea in vaccinees, the consistent and significant observed reduction in disease severity and loose stool output is encouraging and supports the continued development of a subunit, parenterally administered ETEC vaccine. Additional research studies utilizing other potential components of a multivalent vaccine have already been described [35,36,37,38,39], one of which has advanced to a Phase 1 clinical trial [40]. Presuming that the CS6-based component, CssBA, reduces the ETEC-attributable disease upon challenge with a CS6-expressing strain, a conceivable pathway towards the development of a multivalent subunit becomes more apparent. Clearly, additional work on co-manufacturing and formulation would be needed; however, such data would support the advancement of this multivalent parenteral vaccine approach.

## Figures and Tables

**Figure 1 microorganisms-12-00288-f001:**
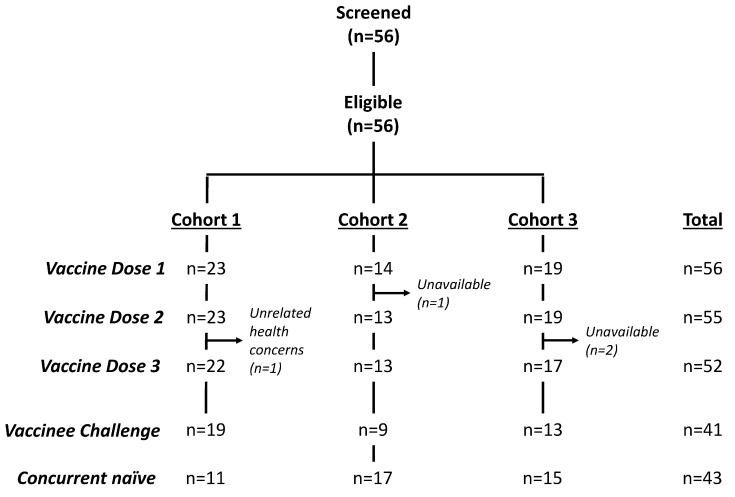
Subject enrollment diagram.

**Figure 2 microorganisms-12-00288-f002:**
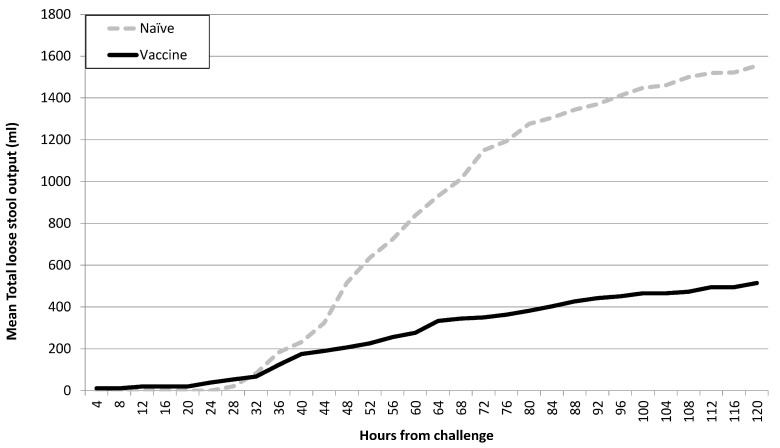
Total stool output (loose; grade 3–5) by vaccination status.

**Figure 3 microorganisms-12-00288-f003:**
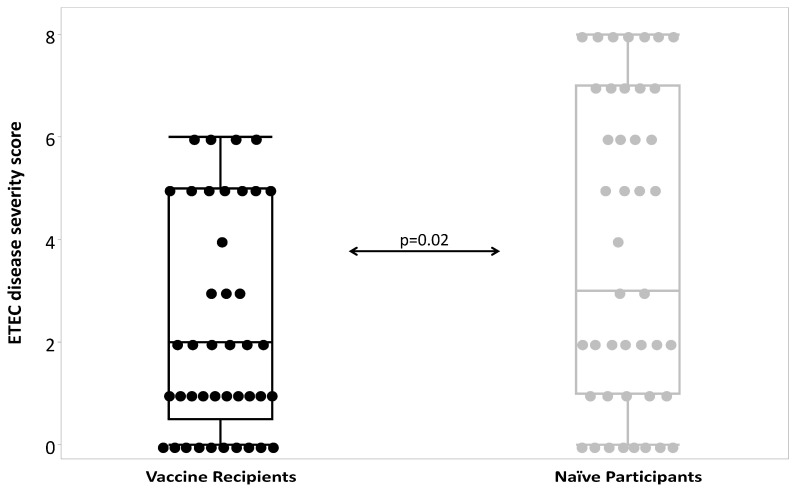
ETEC disease severity scores by vaccination status.

**Figure 4 microorganisms-12-00288-f004:**
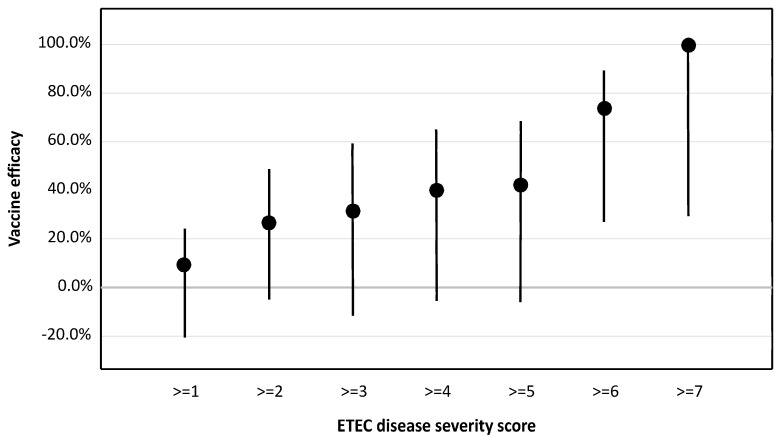
Estimated vaccine efficacy by ETEC disease severity score.

**Figure 5 microorganisms-12-00288-f005:**
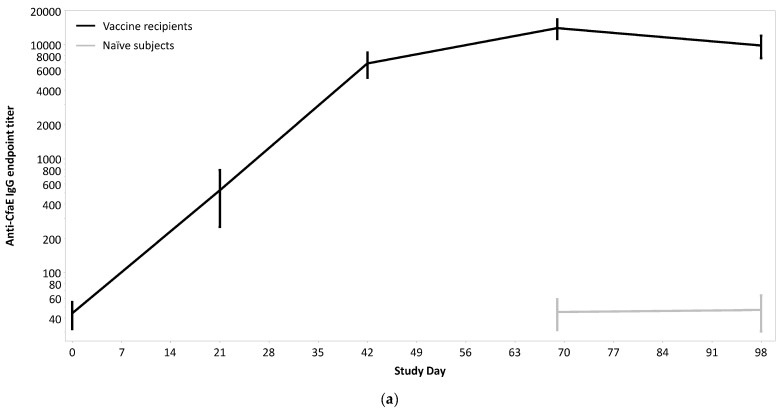
(**a**) Serologic responses (anti-CfaE endpoint titers) after vaccination and/or challenge with H10407; (**b**) serologic response (anti-CfaE IgA endpoint titers) after vaccination and/or challenge with H10407; (**c**) serologic responses (anti-LTB IgG endpoint titers) after vaccination and/or challenge with H10407; (**d**) serologic responses (anti-LTB IgA endpoint titers) after vaccination and/or challenge with H10407.

**Figure 6 microorganisms-12-00288-f006:**
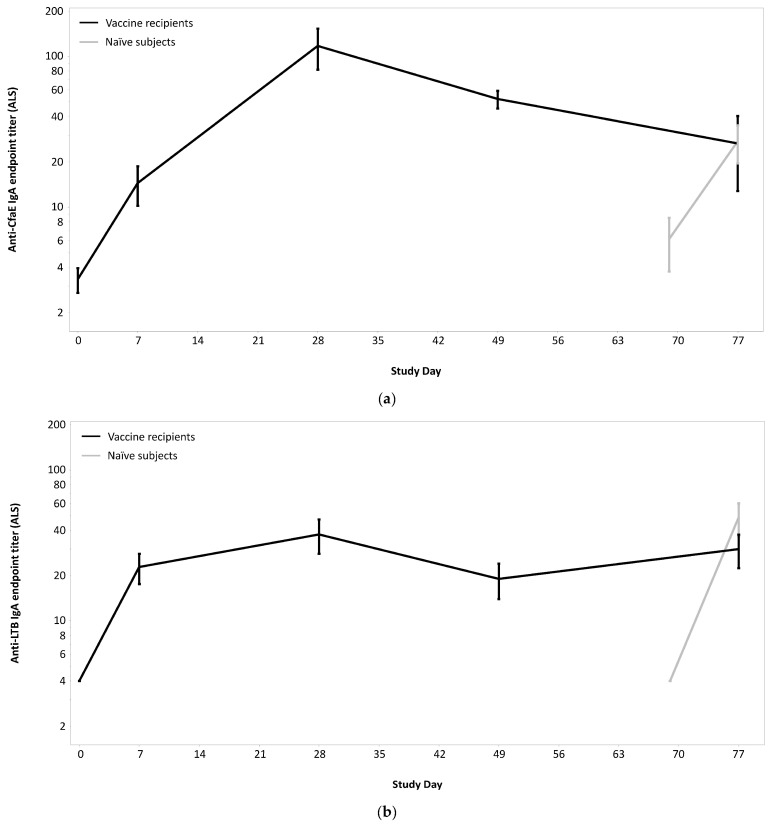
(**a**) Antibody in lymphocyte supernatant responses (anti-CfaE IgA endpoint titers) after vaccination and/or challenge with H10407; (**b**) antibody in lymphocyte supernatant (ALS) responses (anti-LTB IgA endpoint titers) after vaccination and/or challenge with H10407.

**Table 1 microorganisms-12-00288-t001:** Demographic features by cohort.

	Cohort 1		Cohort 2		Cohort 3		All Cohorts	
	Vaccine (*n* = 19)	Naïve (*n* = 11)	Vaccine (*n* = 9)	Naïve (*n* = 17)	Vaccine (*n* = 13)	Naïve (*n* = 15)	Vaccine (*n* = 41)	Naïve (*n* = 43)
**Mean Age (SD)**	34.0 (8.2)	32.6 (9.0)	39.0 (8.4)	36.0 (9.7)	32.5 (8.8)	33.1 (7.1)	34.6 (8.6)	34.1 (8.6)
**Sex; *n* (%)**								
Male	13 (68.4)	8 (72.7)	7 (77.8)	12 (70.6)	8 (61.5)	6 (40.0)	28 (68.3)	26 (60.5)
Female	6 (31.6)	3 (27.3)	2 (22.2)	5 (29.4)	5 (38.5)	9 (60.0)	13 (31.7)	17 (39.5)
**Race; *n* (%)**								
Caucasian	1 (5.3)	1 (9.1)	0 (0.0)	1 (5.9)	1 (7.7)	2 (14.3)	2 (4.9)	4 (9.5)
Black	18 (94.7)	10 (90.9)	8 (88.9)	16 (94.2)	12 (92.3)	9 (64.3)	38 (92.7)	35 (83.3)
Asian	0 (0.0)	0 (0.0)	1 (11.1)	0 (0.0)	0 (0.0)	0 (0.0)	1 (2.4)	0 (0.0)
Multi-Race	0 (0.0)	0 (0.0)	0 (0.0)	0 (0.0)	0 (0.0)	3 (21.4)	0 (0.0)	3 (7.1)
**Ethnicity; *n* (%)**								
Hispanic/Latino	0 (0.0)	0 (0.0)	0 (0.0)	1 (5.9)	2 (15.4)	2 (13.3)	2 (4.9)	3 (7.0)
Non-Hispanic/Latino	19 (100)	11 (100)	9 (100)	16 (94.1)	11 (84.6)	13 (86.7)	39 (95.1)	40 (93.0)

**Table 2 microorganisms-12-00288-t002:** Frequency of adverse events coded as related to the vaccine by cohort.

Adverse Event	Cohort 1 (*N* = 23)	Cohort 2 (*N* = 14)	Cohort 3 (*N* = 19)	Total (*N* = 56)
Decreased Absolute Lymphocytes	1 (4.3)	0 (0.0)	0 (0.0)	1 (1.8)
Elevated ALT	1 (4.3)	0 (0.0)	0 (0.0)	1 (1.8)
Chills	2 (8.7)	1 (7.1)	0 (0.0)	3 (5.4)
Fever	1 (4.3)	0 (0.00)	0 (0.0)	1 (1.8)
Arthralgia	2 (8.7)	2 (14.3)	0 (0.0)	4 (7.1)
Myalgia	2 (8.7)	2 (14.3)	0 (0.0)	4 (7.1)
Lightheadedness	1 (4.3)	0 (0.0)	1 (5.3)	2 (3.6)
Malaise	2 (8.7)	2 (14.3)	0 (0.0)	4 (7.1)
Fatigue	0 (0.0)	0 (0.0)	1 (5.3)	1 (1.8)
Headache	3 (13.0)	4 (28.6)	1 (5.3)	8 (14.3)
Abdominal Pain	1 (4.3)	0 (0.0)	0 (0.0)	1 (1.8)
Nausea	2 (8.7)	0 (0.0)	0 (0.0)	2 (3.6)
Muscle Twitch Right Arm	0 (0.0)	1 (7.1)	0 (0.0)	1 (1.8)
Right Deltoid Pain	0 (0.0)	1 (7.1)	0 (0.0)	1 (1.8)
Left Deltoid Pain	0 (0.0)	1 (7.1)	0 (0.0)	1 (1.8)
Vaccine Site Pain	5 (21.7)	5 (35.7)	1 (5.3)	11 (19.6)
Vaccine Site Pruritus	21 (91.3)	10 (71.4)	18 (94.7)	49 (87.5)
Vaccine Site Reaction	23 (100)	14 (100)	19 (100)	56 (100)
Vaccine Site Swelling	1 (4.3)	0 (0.0)	7 (36.8)	8 (12.5)
Vaccine Site Tenderness	14 (60.9)	6 (42.9)	12 (63.2)	32 (57.1)

**Table 3 microorganisms-12-00288-t003:** Vaccine site appearance grading scale reactions by vaccine dose.

	Dose 1 (*n* = 56)	Dose 2 (*n* = 55)	Dose 3 (*n* = 52)
Erythema (any)	56 (100%)	37 (67%)	36 (69%)
Score 1	15 (27%)	16 (29%)	15 (29%)
Score 2	27 (48%)	13 (24%)	11 (21%)
Score 3	14 (25%)	8 (15%)	10 (19%)
Score 4	0 (0%)	0 (0%)	0 (0%)
Score 5	0 (0%)	0 (0%)	0 (0%)
Induration (any)	56 (100%)	54 (98%)	52 (100%)
Score 1	0 (0%)	0 (0%)	2 (4%)
Score 2	16 (29%)	30 (55%)	35 (67%)
Score 3	29 (52%)	16 (29%)	15 (29%)
Score 4	8 (14%)	5 (9%)	0 (0%)
Score 5	3 (5%)	3 (5%)	0 (0%)
Hyperpigmentation (any)	56 (100%)	55 (100%)	50 (96%)
Score 1	2 (4%)	6 (11%)	11 (21%)
Score 2	9 (16%)	25 (45%)	21 (40%)
Score 3	28 (50%)	16 (29%)	16 (31%)
Score 4	17 (30%)	8 (15%)	2 (4%)
Score 5	0 (0%)	0 (0%)	0 (0%)
Hypopigmentation (any)	1 (2%)	9 (16%)	3 (6%)
Score 1	0 (0%)	4 (7%)	1 (2%)
Score 2	1 (2%)	3 (5%)	1 (2%)
Score 3	0 (0%)	2 (4%)	1 (2%)
Score 4	0 (0%)	0 (0%)	0 (0%)
Score 5	0 (0%)	0 (0%)	0 (0%)
Edema (any)	11 (20%)	0 (0%)	0 (0%)
Score 1	0 (0%)	0 (0%)	0 (0%)
Score 2	8 (14%)	0 (0%)	0 (0%)
Score 3	3 (5%)	0 (0%)	0 (0%)
Score 4	0 (0%)	0 (0%)	0 (0%)
Score 5	0 (0%)	0 (0%)	0 (0%)

Footnote: Plaques (score 2) were only observed at the vaccine site in a single subject. Similarly, vesicles (score 1) were noted on a single subject at the vaccination site.

**Table 4 microorganisms-12-00288-t004:** Percent of subjects meeting the primary endpoint of moderate-to-severe diarrhea (MSD) by cohort and group (vaccinee and naïve).

Cohort	Challenge Dose (cfu)	Naïve *n*/*N* (%)	Vaccinee *n*/*N* (%)	Efficacy Estimate 95% CI (%)
1	1.0 × 10^7^	5/11 (45.5%)	9/19 (47.4%)	−3.6 (−106.2–47.9)
2	1.2 × 10^7^	11/17 (64.7%)	1/9 (11.1%)	60.3 (21.3–80.0)
3	1.9 × 10^7^	8/15 (53.3%)	6/13 (46.2%)	13.3 (−81.5–58.6)
All	--	**24/43 (55.8%)**	**16/41 (39.0%)**	*** 27.8 (−7.5–51.6)**

* Common risk ratio estimated using the Mantel–Haenszel method; Breslow–Day Test for Homogeneity chi-square: 4.7061 (*p* = 0.095).

**Table 5 microorganisms-12-00288-t005:** Frequency [n (%)] of challenge-associated events.

Adverse Event	Naïve			Vaccine		
	Mild	Mod-Sev	Any	Mild	Mod-Sev	Any
Abdominal Cramps	8 (18.6)	20 (46.5)	28 (65.1)	11 (26.8)	6 (14.6)	17 (41.5)
Abdominal Pain	0 (0.0)	15 (34.9)	15 (34.9)	3 (7.3)	4 (9.8)	7 (17.1)
Anorexia	4 (9.3)	19 (44.2)	23 (53.5)	6 (14.6)	6 (14.6)	12 (29.3)
Headache	7 (16.3)	10 (23.3)	17 (39.5)	5 (12.2)	8 (19.5)	13 (31.7)
Lightheadedness	5 (11.6)	3 (7.0)	8 (18.6)	3 (7.3)	1 (2.4)	4 (9.8)
Malaise	4 (9.3)	17 (39.5)	21 (48.8)	7 (17.1)	6 (14.6)	13 (31.7)
Vomiting	4 (9.3)	8 (18.6)	12 (27.9)	2 (4.9)	1 (2.4)	3 (7.3)
Nausea	11 (25.6)	8 (18.6)	19 (44.2)	6 (14.6)	4 (9.8)	10 (24.4)

Footnote: Mod-Sev, moderate to severe.

**Table 6 microorganisms-12-00288-t006:** Number (%) of vaccine recipients developing an immune response prior to challenge.

	Serum IgG	Serum IgA	ALS IgA
anti-CfaE	41 (100)	25 (61.0)	40 (97.6)
anti-LT	36 (87.8)	26 (63.4)	28 (68.3)

Footnote: Responder defined as ≥4-fold rise in baseline titer.

## Data Availability

Data are contained within the article and in associated Tables and Figures. The article will be published in a fully open-access journal to help ensure widespread data dissemination.

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
