# Peer review of "Efficacy Evaluation of an Intradermally Delivered Enterotoxigenic *Escherichia coli* CF Antigen I Fimbrial Tip Adhesin Vaccine Coadministered with Heat-Labile Enterotoxin with LT(R192G) against Experimental Challenge with Enterotoxigenic *E. coli* H10407 in Healthy Adult Volunteers"

_microorganisms, 2024, doi:10.3390/microorganisms12020288_

Round 1
Reviewer 1 Report
Comments and Suggestions for Authors
This study is a clinical trial protocol for a new vaccine against bacterial diarrhoea in humans caused by Enterotoxigenic E. coli (ETEC). The authors point out that the previously achieved results allow to consider this immune composition promising for the development of a vaccine preparation. Thus, the immunogenicity of the vaccine was proved in an experimental mouse model. That is, the authors developed a stable monomeric form of CfaE (dscCfaE), which has high immunogenicity when co-administered with LT-based antigen in Aotus nancymaae mice, with different methods of administration (intranasal and intradermal), as well as in mice by transcutaneous administration. All these prerequisites together with low side effects allowed the authors of the paper to proceed to human trials and approach the first phase of clinical trials on the efficacy of the new vaccine preparation. In their narrative, the authors state that the components of the investigational vaccine, dscCfaE and LTR192G, were manufactured using current good manufacturing practices (cGMP). This is an important point that is necessary to form the final dossier for pharmacovigilance approval.
Statistically and according to the results of the trial, the study can be called a success. The sample of volunteer patients, in general for all sets, did not decrease significantly and remained within the study. Among the 56 subjects who signed written informed consent and underwent specific screening for the vaccination phase, 52 received all 3 doses of the vaccine.
The vaccine safety study showed that there were no serious adverse events associated with the vaccine; moreover, there were no adverse events that led to rejection of the vaccine in the first phases of the clinical trial.
Some of the shortcomings that do not detract from the significance of this serious and comprehensive work can be attributed to some points from the experimental part:
Lines 171-173 state that IgG and IgA assay plates were "washed and ExtrAvidin® - peroxidase (Sigma-Aldrich) was added at a ratio of 1:2000 and incubated for 30 min at room temperature (RT)". Although lines above, the addition of the peroxidase conjugate had already occurred (lines 169-170): "Plates were washed five times with PBS-T followed by addition of 0.5 μg/ml peroxidase-conjugated goat anti-human IgG (KPL, Gaithersburg, MD) or 0.25 μg/ml biotin-conjugated anti-human IgA (KPL) in 1% (for IgG) or 2% (for IgA) skimmed milk-PBS-T for 1.5 h at RT." It is not very clear what the authors wanted to show by this. This snippet needs clarification (why make the conjugate twice?).
Perhaps, for easier reproduction of the immunoenzymatic analysis described in this paper it would be necessary to split into two separate descriptions of Antigen-specific ELISA for IgG and IgA, but this is just a wish on my part for the authors.
In general, the work makes a good impression and can be accepted for publication after minor corrections.
Author Response
Thank you for your thorough evaluation of our paper. We appreciate your positive feedback and acknowledgment of the study's potential.
Regarding your concerns about the experimental section, particularly in the description of peroxidase conjugate addition, we have carefully reviewed the manuscript. After thorough consideration, we regret to that we are unable to implement the changes due to the tight timeline and availability of our immunologist to revise this section.
Wee hope that the overall quality and significance of our work warrant its acceptance for publication.
Reviewer 2 Report
Comments and Suggestions for Authors
Dear Authors and Editors,
I hope this message finds you well. I am writing to bring to your attention some concerns I have encountered while reviewing the manuscript titled "Efficacy Evaluation of an Intradermally Delivered Enterotoxigenic Escherichia coli CFA/I Fimbrial Tip Adhesin Vaccine Co-administered with Heat-Labile Enterotoxin with LT(R192G) against Experimental Challenge with ETEC H10407 in Healthy Adult Volunteers."
1. Figure Presentation:
In the text, there is a reference to Figures 1-6, but unfortunately, these figures are not included in the manuscript. To conduct a thorough review, I kindly request that the complete set of figures be provided.
2. Nomenclature Usage:
Please ensure consistent usage of nomenclature, specifically when introducing the whole name followed by the abbreviation. Additionally, consider italicizing "Escherichia coli" and using "E. coli" in italics as well for uniformity.
3. Missing Information in Results:
It appears that some information is missing in the Results section. I kindly request that the missing details be provided for a comprehensive evaluation.
4. Statistical Significance:
According to my calculations, all results presented in Table 4 are non-significant. I would appreciate a clarification or additional information regarding the statistical significance of the results.

I understand that these points may require your attention, and I appreciate your prompt response to address these concerns. Once the necessary revisions are made, I will be able to conduct a more thorough and accurate assessment of the manuscript.
Thank you for your understanding and cooperation.
Best regards,
Anna

Author Response
We appreciate the review and comments from Reviewer 2. We follow with comments/responses to the concerns raised by the reviewer.
1 & 3. Figure Presentation and Missing Results. It appears the publisher had not embedded submitted Figures in the Manuscript which was prepared for the author. We will request the publisher format and embed Figures 1-6 in the subsequent proof send for additional review. Thank you for the comment and raising this issue.
2. Nomenclature usage. We reviewed the manuscript and ensured italics were used in appropriate instances.
4. Statistical Significance. We appreciate the comments with respect to the level of significance achieved against the primary endpoints for vaccine efficacy displayed on Table 4. We have added additional text in the discussion section to further clarify and describe these observation. We included 95% confidence intervals for each cohort's estimate of efficacy. The ranges for cohort 2 do not cross 0%; however, across all three cohorts, there was heterogeneity that was borderline statistically significant (p=0.095) indicating that the combined efficacy estimate should be interpreted with caution due to the heterogeneity in the efficacy estimates across the three cohorts, which we are still unable to fully explain. In lines 346-347 of the manuscript we state that the "vaccine did not elicit a statistically significant reduction in moderate-to-severe diarrhea rates vaccinated participants compared to contemporaneously challenged naive subjects" and in the following line state we did not meet "our primary efficacy endpoint."
We have added an additional sentence in the Discussion section to clarify these points.
Thank you again for the thorough review.

Reviewer 3 Report
Comments and Suggestions for Authors
The study investigates the protective efficacy of an intradermally administered vaccine containing a subunit of the colonization factor antigen I (CFA/I), specifically CfaE, along with a mutant E. coli heat-labile enterotoxin, LTR192G. This vaccination strategy is assessed in a controlled human infection model (CHIM) involving cohorts of healthy adult subjects.
Three doses of the vaccine were given at 3-week intervals to enrolled subjects, followed by a challenge with a CFA/I+ ETEC strain. The results indicate that the vaccinated group experienced a reduced incidence of moderate to severe diarrhea compared to the unvaccinated group. The overall efficacy estimate was 27.8%, with a significant reduction in loose stool output and overall ETEC disease severity.
The study is notable for being the first to demonstrate protection against ETEC challenge through intradermal vaccination with an ETEC adhesin. However, the variability in naïve attack rates among the cohorts suggests a need for further examination of the challenge methodology used in the study. Overall, the findings suggest promise for this vaccination approach in mitigating the severity of ETEC-related symptoms.the vaccination did demonstrate a significant reduction in overall ETEC disease severity, lower loose stool output, and fewer ETEC-attributable signs and symptoms.
The results, while not meeting the primary efficacy endpoint, align with previous work demonstrating the ability of passively administered bovine colostrum antibodies against CfaE to prevent ETEC-mediated disease. The study emphasizes the importance of considering clinical outcomes beyond diarrhea volume and frequency, especially in capturing the full spectrum and severity of ETEC illness.
The discussion also acknowledges the heterogeneity in diarrhea rates among cohorts, underscoring the challenges associated with enteric CHIMs and the need for continued refinement and standardization. Efforts to identify underlying reasons for variability, including baseline antibody titers, were inconclusive. Despite the absence of a significant reduction in diarrhea rates, the observed consistent reduction in disease severity and stool output supports the ongoing development of a subunit, parenterally administered ETEC vaccine. The potential for a multivalent vaccine, incorporating additional components, is also discussed, paving the way for further research and formulation considerations.
Author Response
We appreciate your insightful review and are grateful for your positive acknowledgment of the study's significance.
We duly note your observation regarding the variability in naïve attack rates among cohorts and understand the importance of further examining the challenge methodology used in our study.
Your recognition of the reduction in overall ETEC disease severity and the discussion on the potential for a multivalent vaccine incorporating additional components are valuable points. We are pleased that the study's findings, despite not meeting the primary efficacy endpoint, contribute to the growing body of knowledge on ETEC vaccination approaches and highlight the importance of considering clinical outcomes beyond diarrhea volume and frequency.
Thank you for the thorough review.
Round 2
Reviewer 2 Report
Comments and Suggestions for Authors
Dear Authors and Editors,
I would like to express my sincere appreciation for your prompt and thorough response to my concerns regarding the manuscript "Efficacy Evaluation of an Intradermally Delivered Enterotoxigenic Escherichia coli CFA/I Fimbrial Tip Adhesin Vaccine Co-administered with Heat-Labile Enterotoxin with LT(R192G) against Experimental Challenge with ETEC H10407 in Healthy Adult Volunteers."
Your attention to detail and commitment to addressing each of the raised points is commendable. It is reassuring to learn that the absence of Figures 1-6 in the manuscript was an issue related to the publisher's formatting, and I appreciate your initiative to rectify this in the subsequent proof. Ensuring the proper presentation of figures is crucial for a comprehensive review, and I am grateful for your commitment to resolving this matter.
Furthermore, your efforts to review and correct the nomenclature usage, specifically italicizing "Escherichia coli" and "E. coli" for uniformity, are noted and appreciated. Consistency in language and formatting enhances the overall professionalism of the manuscript.
I am particularly pleased with the detailed explanation provided regarding the statistical significance of the results in Table 4. Your addition of 95% confidence intervals and the acknowledgment of heterogeneity across cohorts contribute significantly to the clarity and transparency of the study's findings. The inclusion of additional text in the discussion section further strengthens the manuscript's scientific rigor.
In conclusion, I commend your dedication to addressing the concerns raised in my review comprehensively. Your response not only reassures the validity of the research but also demonstrates your commitment to producing a high-quality manuscript.
Best regards,
Anna
Author Response
Good morning,
Thank you for reviewing our revised manuscript and for the kind words. We appreciated the suggested edits and are pleased these were well received and conveyed the nuances of this trial effectively.
We will continue to review further proofs in detail to ensure all data representations are included in the published report.
Thank you again and best wishes.